# Lower Adherence to Lifestyle Recommendations of the World Cancer Research Fund/American Institute for Cancer Research (2018) Is Associated with Decreased Overall 10-Year Survival in Women with Breast Cancer

**DOI:** 10.3390/nu17061001

**Published:** 2025-03-12

**Authors:** Jaqueline Schroeder de Souza, Luiza Kuhnen Reitz, Cândice Laís Knöner Copetti, Yara Maria Franco Moreno, Francilene Gracieli Kunradi Vieira, Patricia Faria Di Pietro

**Affiliations:** 1Post Graduate Program in Nutrition, Federal University of Santa Catarina, Florianopolis 88040-900, Brazil; jaqueline.schroeder04@gmail.com (J.S.d.S.); candice.lk@hotmail.com (C.L.K.C.); yara.moreno@ufsc.br (Y.M.F.M.); francilene.vieira@ufsc.br (F.G.K.V.); 2Florianopolis Specialized Oncology Center, Florianopolis 88032-005, Brazil; luizakreitz@gmail.com

**Keywords:** breast neoplasms, survival, mortality, recurrence, diet

## Abstract

Background/Objectives: The 2018 lifestyle recommendations of the World Cancer Research Fund/American Institute for Cancer Research (WCRF/AICR) have been associated with lower incidence of breast cancer, but their impact on mortality, survival, and recurrence of the disease is not known. This study aimed to investigate the association between adherence to WCRF/AICR recommendations on mortality, overall 10-year survival, and recurrence in women diagnosed with breast cancer. Methods: This prospective study included 101 women diagnosed with breast cancer between 2006 and 2011. Food consumption, body weight, and physical activity data were collected at diagnosis to calculate the WCRF/AICR score. Mortality, survival, and recurrence data were collected in 2020–2021. Kaplan–Meier curves and Cox proportional hazards regression models were applied to verify the association between WCRF/AICR score and its components with outcomes. Results: Women with lower adherence to the WCRF/AICR score (1st tertile, which was the reference group for regression models) had lower chance of 10-year overall survival when compared to patients with higher scores (2nd and 3rd tertiles, n = 67) (*p* = 0.025). Consumption of sugary drinks increased the chance of all-cause mortality (*p* = 0.005) and daily fiber intake increased the chance of overall 10-year survival (*p* = 0.027). Conclusions: It is suggested that adherence to WCRF/AICR recommendations before breast cancer treatment may contribute to better life expectancy.

## 1. Introduction

Breast cancer is the most prevalent neoplasm and the leading cause of cancer death among women in Brazil and worldwide [1,2]. It is estimated that, from 2022 to 2025, there will be 2.45 million new cases of breast cancer [1].

Even after antineoplastic treatment, patients with breast cancer are susceptible to recurrence or the occurrence of a new type of cancer. The chance of having a recurrence is proportionally related to the size of the breast tumor and lymph node involvement [3,4]. Furthermore, it is known that breast carcinogenesis, both primary and recurrent, is closely related to exposure to modifiable factors such as diet, physical activity, and body weight. Given this important relationship, in 2018 the World Cancer Research Fund (WCRF) and the American Institute for Cancer Research (AICR) published a report in which ten recommendations are presented for cancer and recurrence prevention [5]. This report had a previous version published in 2007 [6], and in 2024, the new WCRF/AICR report synthesized the latest evidence on lifestyle factors to provide guidance for individuals living with and beyond breast cancer. Key health components, such as maintaining a healthy body weight and consuming adequate dietary fiber, have been associated with improved health outcomes [7]. It has already been shown that adherence to the WCRF/AICR [5,6] recommendations is inversely associated with the risk of several types of cancer [8,9,10,11], including breast cancer [12,13,14,15,16]. Turati et al. [14] found that 25% of breast cancer cases were related to low to moderate adherence to the 2018 recommendations of the WCRF/AICR. The inverse association between adherence to 2018 recommendations of the WCRF/AICR and risk of breast cancer is mainly observed in postmenopausal women, with luminal clinical subtypes of positive expression for estrogen and progesterone, and obese women [16].

Some biological mechanisms that possibly explain this protective factor against tumor development are the reduction in oxidative stress [17], influenced by the phytochemicals of a diet rich in plant foods and the increase in the number of mitochondria by the practice of physical exercise [17,18], and increased insulin sensitivity, immunity, and reduced inflammation provided by healthy lifestyle habits, aspects that are inversely associated with the emergence of several chronic non-communicable diseases, including cancer [5,19,20]. Adherence to the 2018 recommendations of the WCRF/AICR has also shown a trend toward a lower risk of all-cause mortality in patients with colorectal [21] and pancreatic [11] cancer. According to the Global Cancer Update Programme (CUP Global) summary of evidence grading [22], the relationship between post-diagnosis body fatness and elevated risks of all-cause mortality, breast cancer-specific mortality, and the development of a second primary breast cancer is supported by strong evidence, indicating a probable causative link. Conversely, the relationship between body fatness and breast cancer recurrence, as well as mortality from non-breast cancer-related causes, is backed by limited evidence, suggesting a potential but less certain causal association. Evidence suggesting that recreational physical activity reduces the risk of all-cause mortality and breast cancer-specific mortality is classified as limited but indicative of a potential association [22].

Considering the WCRF/AICR recommendations released in 2007 [6], which present some differences from the updated recommendations [5], such as the cut-off points for the consumption of alcoholic beverages and specific recommendations on limiting the consumption of salt and not consuming moldy cereals and grains, some studies have already shown an inverse relationship between following these guidelines and mortality in breast cancer patients [23,24,25]. However, the investigation of the influence of adherence to updated WCRF/AICR recommendations on survival time, recurrence, and mortality has not yet been carried out in a specific sample of Brazilian breast cancer survivors, which highlights the relevance of the present study, considering the epidemiological impact of this disease.

Based on the fact that proper adherence to these lifestyle guidelines has already been shown to reduce mortality in specific cancer [11,21] and is capable of positively interfering with several health outcomes, such as better quality of life and functional capacity [26], which are related to survival [27,28], it is hypothesized that following the 2018 recommendations of the WCRF/AICR can increase survival, reduce the chance of recurrence and mortality in women with breast cancer. Based on this, the present study aimed to investigate the association between adherence to WCRF/AICR recommendations (global score and specific components score) on mortality, overall 10-year survival, and recurrence in women diagnosed with breast cancer.

## 2. Materials and Methods

This observational study included a sample of patients (n = 101) admitted to the Carmela Dutra Maternity Hospital (Florianopolis, Santa Catarina, Brazil) for breast tumor removal surgery between October 2006 and August 2011. We highlight that our study is classified as prospective and observational, since different types of variables were collected over time. Data collection involved the baseline period, between the years 2006 and 2011, and the investigation of outcomes (overall 10-year survival, mortality, and recurrence), with data collection between 2020 and 2021. The recruitment of women was based on convenience, as adopted in previous studies [29,30,31] and the research exclusion criteria were a clinical history of neoplasia; having undergone surgery within the past year; pregnant or lactating women; benign breast neoplasm; presence of neurological diseases or acquired immunodeficiency virus; undergoing neoadjuvant treatment; and lack of data on outcomes of the study. The sampling process is described in item Outcomes.

### 2.1. Baseline

#### 2.1.1. Data on Sociodemographic, Anthropometric, Clinical, and Dietary Aspects

At baseline, patients had malignancy of the disease confirmed by anatomopathological examination and were interviewed before the surgical treatment. The classification of breast cancer stage was based on the TNM system [32]. Sociodemographic, clinical, anthropometric, and food consumption data were collected before adjuvant treatment. Comorbidities were also investigated at baseline. Data on the detailing of adjuvant treatment were collected after the last chemotherapy or radiotherapy session (mean duration of adjuvant treatment: 11.2 months, standard deviation—SD: 16.7). To collect these data, before breast cancer surgery, trained interviewers applied the questionnaire developed by Vieira et al. [33] and adapted by Rockenbach [34]. After adjuvant treatment, data were collected on the type of antineoplastic treatment performed and the number of chemotherapy and radiotherapy sessions [34].

Anthropometric data collected before adjuvant treatment (baseline) were current body weight (kg), height (m), and waist circumference (cm). For current body weight and height, an anthropometric scale with an attached stadiometer was used (Filizola^®^ brand, São Paulo—Brazil, capacity of 150 kg; precision of 100 g). Protocolled anthropometric techniques [35,36] were adopted to measure these parameters, and Body Mass Index was calculated [36,37]. An inelastic anthropometric tape (Cescorf Inc., Porto Alegre, Rio Grande do Sul) with an accuracy of 0.1 cm was used to measure the patient’s waist circumference, and the classification proposed by the World Health Organization for this indicator was considered [37].

The assessment of current physical activity was also carried out at the research baseline. Participant was asked about the weekly frequency, duration, and type of physical exercise usually performed, which allowed the classification according to the U.S. Physical Activity Guidelines [38].

Food consumption data were obtained by applying a validated Food Frequency Questionnaire (FFQ) containing 112 food items [39,40], which refers to the usual diet of the previous year. Food consumption frequencies were converted into daily amounts consumed [41], allowing the analysis of the nutritional composition of the participants’ diet [42,43]. For the evaluation of the fiber intake, the amount of this component was adjusted by energy [44]. The seasonality of some foods, such as fruits and vegetables, was considered according to the harvest table of the Secretariat of Agriculture and Supply of the State of São Paulo to estimate daily consumption [45].

#### 2.1.2. WCRF/AICR Score

The adherence to WCRF/AICR recommendations was accessed according to a scoring system which ranges from 0 to 7 points, proposed by Shams-White et al. [46]. Each recommendation originates components with specific scores, and the sum of them provides the total score, which is proportional to the follow-up of the recommendations [46], according to Table 1. It is important to highlight that, as this is an observational study, the women involved in the research were not subjected to any type of intervention to increase their knowledge of the WCRF/AICR recommendations. The researchers’ intention was to relate the lifestyle framework with recommendations published after the baseline data collection. The participants’ lifestyle was not reassessed in the second stage of data collection (outcomes data collection), since the aim of the research was to evaluate the influence of lifestyle at the time of breast cancer diagnosis on clinical outcomes of mortality, survival and recurrence. It is well established that pre-diagnosis lifestyle habits can influence prognosis [47], which was the primary research focus of the investigators in this study.

The calculation of the WCRF/AICR score was carried out in 2021, that is, after data collection (baseline that occurred between 2006 and 2011). The research team had access to all the necessary data to enable the calculation of the WCRF/AICR score, without requiring the collection of any additional data for operationalization.

We would also like to highlight that the most recent abbreviated version of the score [50] has already been released. However, we chose to use the proposal by Shams-White et al. [46], as it incorporates more components of the WCRF/AICR guidelines.

### 2.2. Outcomes

#### Data on Mortality, Survival, and Recurrence

From December 2020 to May 2021, women and/or family members were contacted to collect information on mortality, recurrence, and survival. Figure 1 demonstrates the sampling process applied in the study.

Figure 2 shows the data collection process regarding mortality, survival, and recurrence. The main form of access to this information was the investigation of clinical records of the Oncological Research Center. In cases where the data in the medical records were insufficient for the research, other data collection methods were used.

The term mortality in this study involved all-cause mortality and specific mortality from breast cancer, defined by the International Classification of Diseases (ICD)—ICD-50 [51], whether accompanied by metastasis and/or other comorbidities. Dead individuals were censored on the date of death [52].

Locoregional recurrence was considered if the cancer recurred in the same location or close to the location (lymph nodes) originally identified in the breast at the first diagnosis. “Metastasis” was the term given to the recurrence of cancer in a location other than the first diagnosis, that is, in another part of the body other than the breast [53,54]. Recurrence within 10 years was considered for participants who presented locoregional recurrence and/or metastasis within 10 years after the first diagnosis of breast cancer (which was confirmed by pathological examination of the surgical specimen).

Regarding survival time, for women who died, the interval between the date of the first diagnosis of breast cancer (date of confirmation of breast cancer by the anatomopathological examination) and the date of death was considered. Among individuals in the sample who did not die, the survival time considered was the time interval between the date of the first diagnosis of breast cancer and the date of the outcomes data collection, as this date was standardized for living patients. The overall 10-year survival was considered for women who did not die within 10 years of the first diagnosis of breast cancer. The term “10-year recurrence” refers to recurrence within 10 years of the primary diagnosis of breast cancer.

For phone calls to patients and/or family members, trained professionals followed a protocol developed by the researchers (Appendix A). The first part of the protocol (10 questions) was aimed at approaching live patients; the second part of the protocol (11 questions) was used for individuals in the sample who died and, in these cases, communication was carried out with the patient’s relatives (Appendix A). Survival time was calculated and tabulated after data collection according to the parameters described in this topic.

### 2.3. Statistical Analyses

For statistical analyses, the score of adherence to the WCRF/AICR recommendations was analyzed in tertiles. For associations, the WCRF/AICR score was analyzed in tertiles: the 1st tertile (reference group for regression models, which represents the group with the lowest adherence to the WCRF/AICR recommendations) and the 2nd with the 3rd tertile, as the focus of the study was to assess the impact of low WCRF/AICR scores on patients’ clinical outcomes.

For the descriptive table, Fisher’s exact test was applied for qualitative variables and Student’s *t*-test for continuous variables. To evaluate the association between the WCRF/AICR score and the outcomes, Kaplan–Meier survival curves were applied. The log-rank test was conducted to compare the two curves (1st WCRF/AICR score tertile versus 2nd and 3rd score tertiles). Time-to-event analysis was assessed by crude Cox proportional hazards regression model and adjusted for age, education (*p* ≤ 0.2), family history of cancer (*p* ≤ 0.2), stage of first cancer, and recurrence (this last variable was excluded when the outcome variable was recurrence and 10-year recurrence). These last two adjustment variables were considered due to the biological plausibility related to the outcomes of interest in this study. Cox proportional hazards regression models were built according to the global WCRF/AICR score and according to the specific components and were conducted with the same adjustment variables. For specific components, the absolute value of each component was considered for the analyses: for example, BMI value, waist circumference, and daily weight of consumption of sugary drinks, among others. All women in the study, including those who died and those who remained alive, were included in all analysis models. Also, all clinical outcomes were modeled as time-dependent (years) covariates in the Cox proportional hazards regression models. From these models, hazard ratios (HR) and respective 95% confidence intervals (CIs) were identified. In all statistical analyses, the significance level considered was *p* < 0.05.

### 2.4. Ethical Approval

The present study followed the ethical principles of research and was approved by the Ethics Committee for Research with Human Beings of the Federal University of Santa Catarina (099/2008, 492/2009 and 4.536.968), the Ethics Committee of the Oncological Research Center (015/2009, 4.639.465, 5.148.664), and the Ethics Committee of the Carmela Dutra Maternity Hospital (0012.0.233.242-10). The study was only conducted with patients (or family members, in the case of deceased patients) who signed an online version of the Free and Informed Consent Form, as established in Resolution 466/2012 of the National Health Council [55].

The Free and Informed Consent Form was sent in the format of an online form (Google Forms) to patients and/or their families via a link provided via WhatsApp and/or email. It should be noted that the Ethics Committees were consulted regarding sending the Free and Informed Consent Form in digital format and approved this file format due to the COVID-19 pandemic. This form provided the necessary guidance for an adequate understanding of the research and was completed digitally by the participants and/or their families who were contacted via telephone call. A copy of the digitally completed Free and Informed Consent Form was sent to the email address of the form responder. Only individuals contacted by telephone who indicated in this document that they agreed to participate in the research took part in the study. The remaining women in the sample who were not contacted by telephone, and who already had all the necessary information in their physical and electronic medical records, did not require sending and signing an informed consent form, since data were obtained via clinical records.

## 3. Results

The sample (n = 101) included women with a mean age of 51 years (SD = 11.3). The studied sample’s average WCRF/AICR score was 3.6 points (SD = 0.9). Regarding the collection of data on mortality, overall survival, and recurrence, for 67.3% of the patients (n = 68) the information was obtained through access to the clinical records, exclusively, data from 31 women (30.8%) were obtained from clinical medical records and also by telephone calls to patients and/or their relatives, and data from 2 women (1.9%) were obtained by contact of the researchers with the Health Centers to which the patients were being followed up.

Table 2 shows sociodemographic and clinical characteristics between the WCRF/AICR score tertiles. A higher proportion of non-white women was found in the group with less adherence to the WCRF/AICR recommendations, that is, the 1st tertile group (*p* = 0.006). The most common comorbidities were systemic arterial hypertension (n = 33, 32.7%), liver disease (n = 16, 15.8%), and diabetes mellitus (n = 15, 14.8%). The most common cause of death was the complication of breast cancer associated with metastasis (involvement of several different organs) (n = 9, 37.5%), isolated metastasis (malignant tumor in a single site other than the breast) (n = 4, 16.7%), and local aggravation of breast cancer (n = 3, 12.5%). The sites that showed the highest percentages of metastasis were the bones (n = 12, 41.4%), lungs (n = 7, 24.1%), and liver (n = 7, 24.1%) (frequencies calculated according to all metastatic sites).

Among patients who had breast cancer recurrence (35.6%, n = 36), 13 women (36.1%) underwent surgical treatment to remove the tumor. Adjuvant treatment for recurrence was observed in 28 women (77.7%), which most often performed was isolated chemotherapy (n = 9, 32.1%), followed by the combination of chemotherapy, radiotherapy, and hormone therapy (n = 5, 17.9%) and chemotherapy associated with hormone therapy (n = 4, 14.3%). The mean time for chemotherapy in cases of recurrence was 10 months (SD = 16), 12 months for radiotherapy (SD = 17.6), and 25 months for hormone therapy (SD = 27.2).

The distribution of Kaplan–Meier curves and the results of the log-rank test (Figure 3, Figure 4, Figure 5, Figure 6 and Figure 7) show a significant difference between groups (1st tertile versus 2nd and 3rd tertiles) for all clinical outcomes in relation to survival time, except for the analysis of 10-year recurrence.

Analysis of patient clinical outcomes indicated that there was an overall all-cause mortality rate of 23.7%, and breast cancer-specific mortality rate of 14.8% (Table 3). The overall 10-year survival, recurrence, and 10-year recurrence rates were 83.1%, 35.6%, and 23.7%, respectively (Table 3). The associations between WCRF/AICR score tertiles and clinical outcomes evaluated by Cox proportional hazards regression models (Table 3) showed that patients in the 1st tertile of WCRF/AICR adherence were less likely to be alive at 10 years after the first diagnosis of breast cancer when compared to patients in the 2nd and 3rd score tertiles (HR = 0.16, 95% CI 0.03–0.8; unadjusted analysis: *p* = 0.025).

Considering the Cox proportional hazards regression models on the specific components of the WCRF/AICR score associated with the outcomes (Appendix A), in the crude models (unadjusted analysis, Appendix A) it was observed that the consumption of sugary drinks increased the chance of all-cause mortality (HR = 1.01, 95% CI 1.01–1.02, *p* = 0.005) and daily fiber intake increased the chance of overall 10-year survival (HR = 1.00, 95% CI 1.01–1.02, *p* = 0.027). In the models that considered the adjustment variables age, education, stage of first cancer, and family history of cancer (Appendix A), it was found that the consumption of sugary drinks still increased the chance of overall mortality (HR = 1.01, 95% CI 1.01–1.02, *p* = 0.026) and daily consumption of fruits and vegetables appeared to be a protective factor for breast cancer-specific mortality (HR = 0.97, 95% CI 0.95–0.99, *p* = 0.022). In the adjusted models that considered the recurrence variable as an adjustment in addition to the other variables mentioned above (Appendix A), daily fiber intake has been shown to contribute to 10-year overall survival (HR = 1.01, 95% CI 1.01–1.02, *p* = 0.022). No significant association was found in the associations of isolated components with recurrence and 10-year recurrence outcomes by crude and adjusted models (Appendix A).

## 4. Discussion

To our knowledge, this is the first study regarding the association between adherence to WCRF/AICR recommendations and mortality, overall 10-year survival, and recurrence in Brazilian women with breast cancer. Herein, women exhibiting lower adherence to the WCRF/AICR recommendations had a lower chance of overall survival at 10 years when compared to patients with higher WCRF/AICR adherence scores, confirming the pre-established hypothesis. This important result demonstrates the relevance of following the WCRF/AICR recommendations [5] for better life expectancy after a breast cancer diagnosis. Although the global WCRF/AICR score relationships with all-cause mortality and breast cancer-specific mortality were not significant in this research, interesting results were found regarding specific components related to mortality and overall 10-year survival. Some studies have already verified an inverse association between the global WCRF/AICR score [6] and mortality after diagnosis of breast cancer [23,24,25]. A longitudinal analysis showed that each 1-point increase in the 2018 WCRF/AICR score was associated with a 9–26% reduction in mortality risk for all outcomes in fully adjusted models, except for the risk of all cancers for male smokers [56].

Many components of the 2018 WCRF/AICR score, mainly related to body weight, diet and physical activity, are associated with a better prognosis for cancer patients [57,58,59]. In our study, we demonstrate that consumption of sugary drinks was associated with an increased chance of all-cause mortality. The systematic review with meta-analysis by Li et al. [60] also corroborates this result on consumption of sugary drinks and mortality and reinforces that the risk of dying is increased mainly due to the development of cardio-vascular diseases and cancer. Other important findings of the present study consist of the relationship between daily fiber intake and increased chance of overall survival in 10 years and the consumption of fruits and vegetables being associated with a protective effect against breast cancer-specific mortality. Regarding diet, De Cicco et al. [61] point out in a literature review that a dietary pattern characterized by high intake of fiber (preferably 30 g/day) from dietary sources such as fruits, vegetables, and whole grains can improve the survival of patients diagnosed with breast cancer. In an Italian cohort [62], following the Mediterranean diet, characterized by high consumption of whole grains and foods of plant origin and limited intake of red meat, has already been shown to be associated with a better prognosis in breast cancer patients. Survival at 15 years was 63.1% for patients with high adherence to a healthy diet, compared to the group with low adherence (53.6%, *p* = 0.013) [62].

Other specific components of the WCRF/AICR (2018) score, despite not being significantly related to the outcomes in this study, also deserve to be highlighted due to their relationship with the prognosis of cancer patients, which is already recognized in the literature. In the umbrella review by Huang et al. [63], the analyses revealed that the increase in the daily consumption of 100 g of red meat and the increase of 50 g/d in the consumption of processed meat increased the risk from 11% to 51% and from 8% to 72%, respectively, of developing multiple types of cancer, as well as being related to mortality from this disease. Thus, it appears that the consumption of red and processed meat does not seem to favor the clinical outcomes of cancer patients [63]. Regarding nutritional status, another component of the WCRF/AICR (2018) score, Sun et al. [57], in a retrospective analysis of 1017 breast cancer patients, found that overweight and obese patients had lower disease-free survival and overall 5-year survival (*p* < 0.001), and higher risks of recurrence and death (*p* < 0.001) than patients with healthy body weight. When it comes to physical activity, a meta-analysis showed that women with breast cancer which were more physically active had a lower risk of all-cause mortality (HR = 0.58, 95% CI: 0.45–0.75) and specific mortality from breast cancer (HR = 0.60, 95% CI: 0.36–0.99) [58]. Concerning alcohol consumption, the results on the association of consumption of this substance with mortality are still controversial [64,65,66]. However, it is already well understood that no dose of alcohol is safe, since ethanol consumption is associated with several health problems, such as cardiovascular diseases, liver cirrhosis, and cancer development [5].

Regarding recurrence, the findings of the current study are similar to the results of other studies, which showed that the associations between following the WCRF/AICR recommendations and cancer recurrence are inconclusive [21,67]. On the other hand, the study by Kaledkiewicz and Szostak-Wegierek [68] showed that breast cancer patients with recurrence had higher BMI values and higher percentages of abdominal fat than patients who did not have recurrence. It is known that excess body weight is closely related to cancer recurrence and compromises quality of life by exposing the individual to different comorbidities [69]. It is worth noting that maintaining healthy body weight is one of the recommendations that the WCRF/AICR advocates [5]. In addition, it is already well established that the inadequate lifestyle, represented by the Western dietary pattern and sedentary lifestyle, promotes high concentrations of pro-inflammatory cytokines that facilitate the development of recurrence and metastases [70].

It is known that following a healthy lifestyle is related to reducing the risk of several types of cancers, such as breast, bladder, colon, esophagus, and liver cancer [11], which leads to an understanding that the risk of recurrence also may decrease. Although more studies, mainly interventional, are needed to gather new scientific evidence on the relationship between adherence to the 2018 recommendations of the WCRF/AICR and recurrence, the WCRF/AICR [5] report itself states that, after the diagnosis of cancer, disease survivors are at increased risk of new cancers and chronic non-communicable diseases, which justifies the relevance of following cancer prevention recommendations even after having the disease.

When addressing the prognosis of patients with breast cancer, it is important to consider that there are factors not associated with lifestyle that can also influence survival. In hormone receptor-negative, HER2-positive breast cancer, achieving a pathologic complete response (pCR) after neoadjuvant therapy is strongly associated with improved survival outcomes, including higher overall survival and event-free survival rates. Studies have shown that patients who attain pCR have a significantly lower risk of recurrence and distant metastases compared to those with residual disease [71,72]. A meta-analysis by Cortazar et al. [71] demonstrated that pCR is a robust surrogate marker for long-term survival, particularly in aggressive breast cancer subtypes, such as HER2-positive and triple-negative breast cancers.

This study provides valuable insights on investigation of the association between adherence to WCRF/AICR recommendations on clinical outcomes in women diagnosed with breast cancer, reflecting the research team’s clinical practice and research experience. However, certain methodological limitations were considered. Although the study is based on data from clinical practice, its design was structured with scientific rigor, including: specific inclusion and exclusion criteria; application of robust statistical methods, such as log-rank test and Cox proportional hazards regression models; and the evaluation of the clinical outcomes of women after 10 years of breast cancer diagnosis, which allowed us to analyze the influence of following the 2018 recommendations of the WCRF/AICR in the long term. This methodology made it possible to mitigate potential biases and increase the relevance of the results.

We acknowledge that the observational nature of the study may limit the generalizability of the results and causal inference. Additionally, data collection in a clinical setting may be subject to information bias and unmeasured confounders. While these limitations should be considered when interpreting the findings, we emphasize that integrating healthcare practice with scientific methods provides a realistic and relevant perspective for applying the results in clinical decision-making.

We also recognize the difficulty in accessing the information of many patients who had to be excluded from the sample. This was partly due to the long period of patient follow-up, which resulted in a reduction in the research sample size. On the other hand, different resources were used to access patients’ clinical data, such as electronic and physical medical records and telephone calls. Another limitation concerns the actual FFQ used, an instrument that is already known to depend on patients’ memory, as it includes a closed list of foods, not allowing mention of foods that are not included in the tool; is less precise in relation to the quantification of the portions usually consumed; and is difficult to apply in elderly individuals and/or with low education. To mitigate these FFQ-related limitations, the researchers involved in the collection of food consumption data underwent training to obtain this information. It should be noted that the FFQ is a recognized instrument to quantify the individual’s usual consumption and not the punctual food consumption of a short period of time. The assessment of physical activity also has limitations, such as the dependence on participants’ self-reports and the sample’s difficulty in classifying the intensity of physical exercise, which can generate inconsistencies in responses. However, the team of interviewers was trained to explain the concept of physical activity and exemplify types of physical exercises and their respective intensities to help the participants. Another limitation of the study is related to the lack of important components in the WCRF/AICR score, such as inadequate sun exposure and smoking, known risk factors for the carcinogenesis process. Despite this, the score adopted in this study was standardized and its application is encouraged by WCRF/AICR to allow comparability of results with other studies [46]. Yet, we recognize the heterogeneity of factors that can influence the participant’s prognosis. However, various potential confounding variables based on biological plausibility were assessed for inclusion in the Cox proportional hazards regression models, and all those with *p* ≤ 0.2 were incorporated into the statistical analyses. We also recognize that it is possible that the participants changed their lifestyle habits over the years of the study; however, the research focus consisted of assessing lifestyle at the baseline of clinical outcomes of interest to the research group, since lifestyle habits prior to treatment can impact the prognosis of women previously diagnosed with breast cancer [47].

The importance of the study is highlighted by the inclusion of primary outcomes of recurrence and survival, variables that, to the best of our knowledge, have not yet been studied in research involving the updated WCRF/AICR recommendations [5] in Brazilian women previously diagnosed with breast cancer. Furthermore, the relevance of this study in the epidemiological context is highlighted, since breast cancer is the main cause of cancer death in women worldwide.

Future research, including longitudinal studies and systematic reviews with meta-analysis, may further explore and validate our findings, offering a deeper understanding of the relationship between adherence to WCRF/AICR recommendations and mortality, overall 10-year survival, and recurrence in women diagnosed with breast cancer.

## 5. Conclusions

The relationship found between lower adherence to the 2018 recommendations of the WCRF/AICR and lower chance of overall survival after breast cancer diagnosis reinforces the importance of following cancer and recurrence prevention guidelines recommended by the WCRF/AICR for women diagnosed with breast cancer. Specific score components, such as consumption of sugary drinks and daily fiber intake, which comes mainly from food sources such as fruits and vegetables, seem to interfere with all-cause mortality and overall 10-year survival, respectively, which reflects the need to strengthen nutritional care for individuals without the disease, since lifestyle habits were assessed at the time of diagnosis, and for cancer survivors. The approach to several lifestyle components that are addressed by the WCRF/AICR score, allowing a global assessment, demonstrates the relevance of applying this method in new studies. In this sense, greater dissemination of WCRF/AICR recommendations to society in general is necessary to guide the population regarding healthy lifestyle habits and preventive action against cancer and/or recurrence of the disease. The elaboration of lectures and informative materials on the subject are relevant for the population to have greater knowledge of the WCRF/AICR recommendations.

## Figures and Tables

**Figure 1 nutrients-17-01001-f001:**
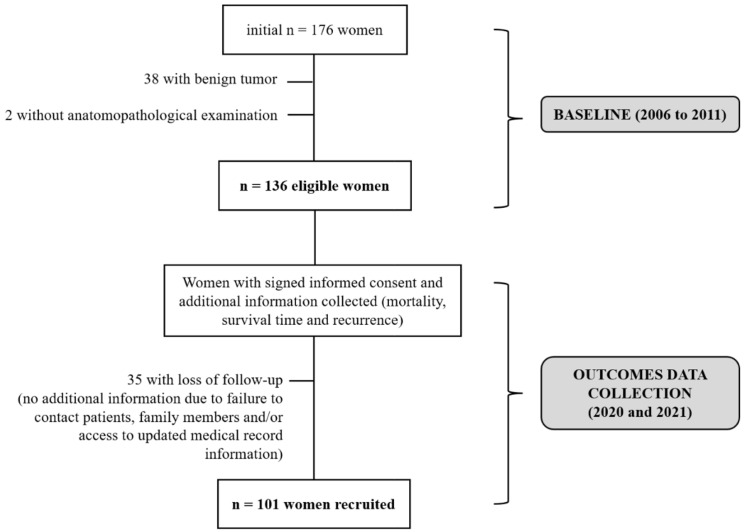
Flowchart of the sampling process of breast cancer patients in the study. Brazil, 2025.

**Figure 2 nutrients-17-01001-f002:**
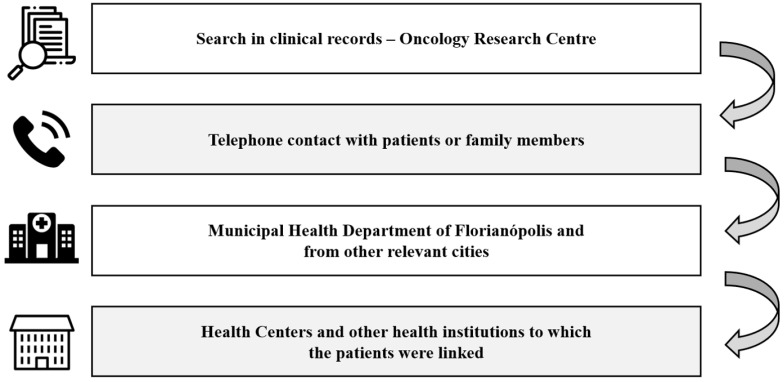
Flowchart of data collection on mortality, survival and recurrence. Brazil, 2025.

**Figure 3 nutrients-17-01001-f003:**
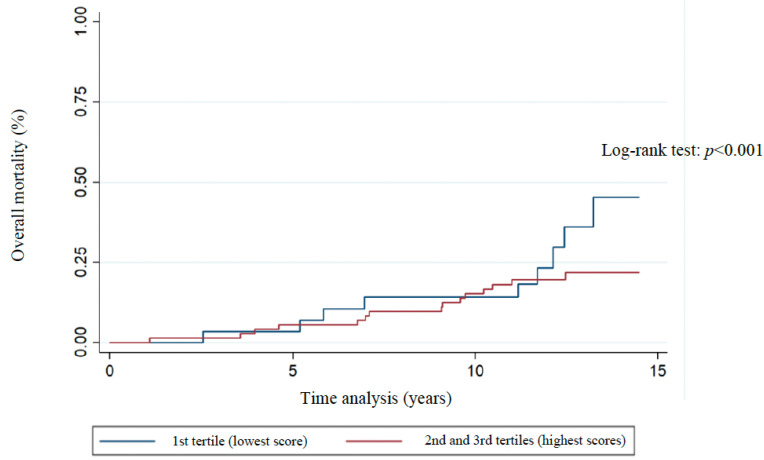
Association of overall mortality with WCRF/AICR score tertiles of women previously diagnosed with breast cancer. Florianopolis, Santa Catarina, Brazil, 2025.

**Figure 4 nutrients-17-01001-f004:**
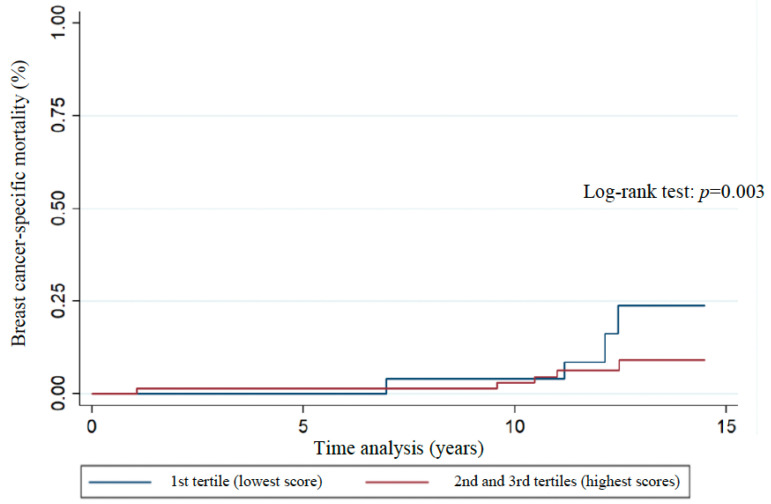
Association of breast cancer-specific mortality with WCRF/AICR score tertiles of women previously diagnosed with breast cancer. Florianopolis, Santa Catarina, Brazil, 2025.

**Figure 5 nutrients-17-01001-f005:**
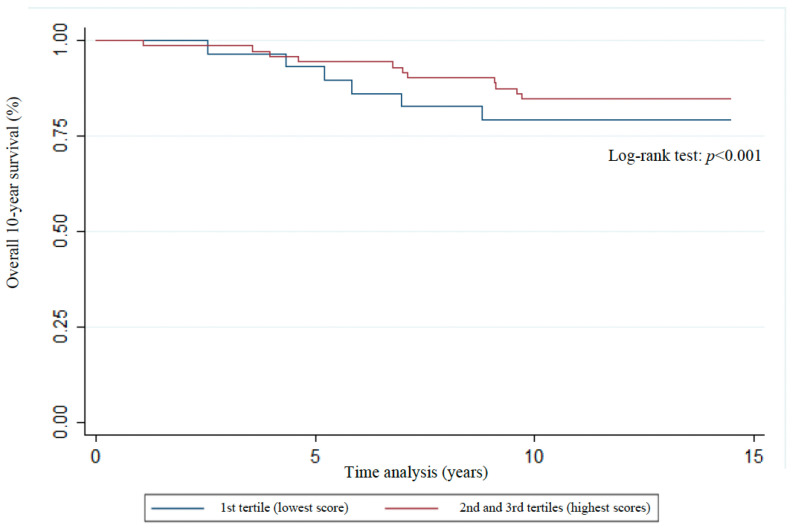
Association of overall 10-year survival^a^ with WCRF/AICR score tertiles of women previously diagnosed with breast cancer. Florianopolis, Santa Catarina, Brazil, 2025. ^a^ Overall 10-year survival refers to women who did not die within 10 years of the first diagnosis of breast cancer.

**Figure 6 nutrients-17-01001-f006:**
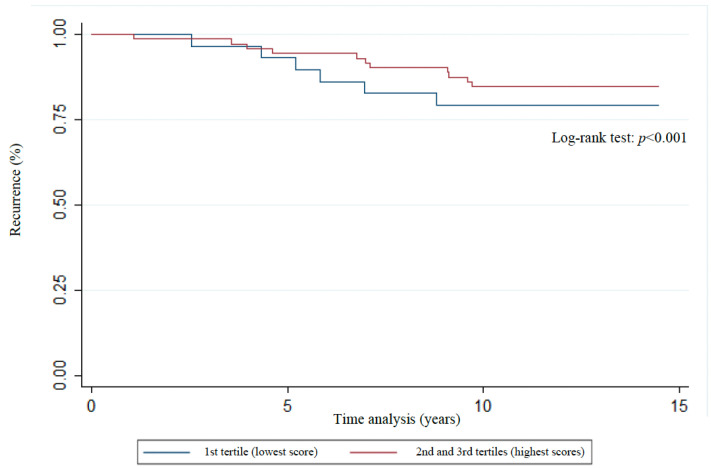
Association of recurrence with WCRF/AICR score tertiles of women previously diagnosed with breast cancer. Florianopolis, Santa Catarina, Brazil, 2025.

**Figure 7 nutrients-17-01001-f007:**
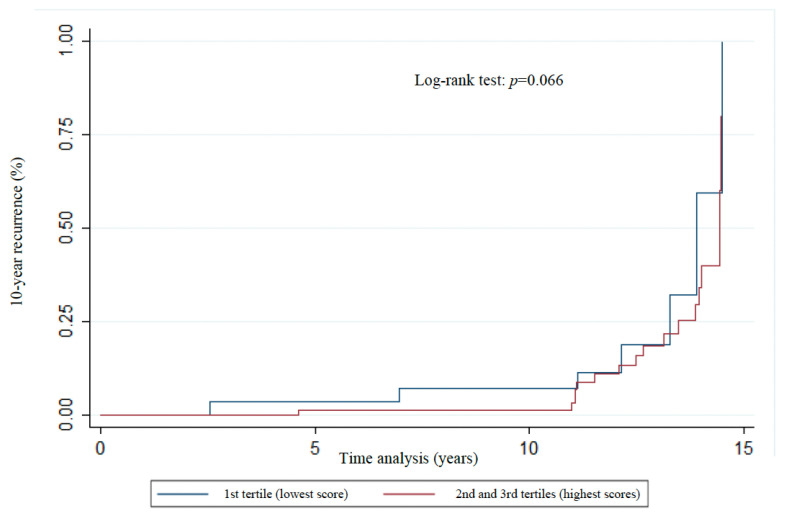
Association of 10-year recurrence ^a^ with WCRF/AICR score tertiles of women previously diagnosed with breast cancer. Florianopolis, Santa Catarina, Brazil, 2025. ^a^ The term “10-year recurrence” refers to recurrence within 10 years of the primary diagnosis of breast cancer.

**Table 1 nutrients-17-01001-t001:** Recommendations of the WCRF/AICR (2018) and its operationalization in an observational study involving women with breast cancer.

WCRF/AICR Recommendations ^1^	Operationalization ^2^	Points ^2^	Operational Notes
	BMI (kg/m^2^)		BMI and waist circumference were classified.
1. Have a healthy body weight	18.5–24.9	0.5	
	25–29.9	0.25	
	<18.5 or 30	0	
	Waist circumference (cm)		
	<80	0.5	
	80–<88	0.25	
	≥88	0	
2. Be physically active	Moderate-vigorous total physical activity (min/week)		According to the duration (min/week). Moderate-intensity exercise: walking, housework, cycling, dancing, and gardening. Vigorous exercise: encompasses fast swimming and cycling, running, aerobic activities
	≥150	1	
	75–<150	0.5	
	<75	0	
3. Eat a diet rich in whole grains, vegetables, fruits and legumes	Fruits and vegetables (g/d)		Analysis of consumption (g/day) of fruits and vegetables and fiber intake. All foods consumed that provide fiber to the diet were considered for fiber intake ^2^
	≥400	0.5	
	200–<400	0.25	
	<200	0	
	Total fibers (g/d)		
	≥30	0.5	
	15–<30	0.25	
	<15	0	
4. Limit consumption of fast food and other processed foods rich in fat, starch, and sugar	Percentage of total kcal of ultra-processed foods		Total energy intake was analyzed in tertiles, and the total energy value of ultra-processed foods consumed was allocated to the specific tertile. Food items characterized as ultra-processed followed the NOVA classification ^3^
5. Limit the consumption of red and processed meat	Total red meat (g/week) andprocessed meat (g/week)		The food items analyzed in this category also followed the NOVA classification ^3^, with beef, pork, and liver being red meat and processed meat attributed to sausage, ham, hamburger, ground beef, and bacon. Such food items were analyzed according to weekly consumption (g/week) ^2^
	Red meat < 500 and processed meat < 21	1	
	Red meat < 500 and processed meat 21–<100	0.5	
	Red meat > 500 or processed meat ≥ 100	0	
6. Limit the consumption of sugary drinks	Total sugary drinks (g/d)		Analysis of the daily consumption of industrialized/artificial juices and soft drinks, refined sugar, and honey ^3^
	0	1	
	>0–≤250	0.5	
	>250	0	
7. Limit alcohol consumption	Total ethanol (g/d)		A conversion of the volume (mL) of alcohol ingested (from any alcoholic beverages, such as wine, beer, and distilled drinks) was carried out in grams of ethanol according to the specific alcohol content of the drink ^4^
	0	1	
	≤14 (1 drink)	0.5	
	>14 (1 drink)	0	
8. Breastfeeding	-	-	It was excluded from the score applied in this study, given that the data necessary for its operationalization were not available. It is noteworthy that the score used allows the total sum without this component ^2^
TOTAL SCORE RANGE		0–7	

BMI—Body Mass Index; WCRF—World Cancer Research Fund; AICR—American Institute for Cancer Research. ^1^ WCRF/AICR, 2018. ^2^ Shams-White et al. [46]. ^3^ Monteiro et al. [48]. ^4^ NIAAA; NCBI, [49]. Table adapted from Shams-White et al. [46].

**Table 2 nutrients-17-01001-t002:** Sociodemographic and clinical characteristics of women with breast cancer according to the score tertiles of adherence to the WCRF/AICR recommendations. Florianopolis, Santa Catarina, Brazil, 2025.

Variable	Groups of Adherence to the WCRF/AICR Recommendations		
	1st Tertile * (n = 34)	2nd and 3rd Tertiles (n = 67)	*p*
**Age (average), SD**	51.1 (1.7)	51.1 (1.4)	0.976 **^a^**
**Number of comorbidities, average (SD)**	2.3 (0.4)	2.4 (0.2)	0.961 **^a^**
**Ethnic group, n (%)**			**0.006 ^b^**
Caucasian	28 (29.8)	66 (70.2)	
Non-white (or Afro-descendant)	6 (85.7)	1 (14.3)	
**Years of study, mean (SD)**	7.6 (0.7)	6.8 (0.5)	0.368 **^a^**
**Marital status, n (%)**			
Married/stable union	21 (31.3)	46 (68.7)	0.511 ^b^
Not married/no stable union	13 (38.2)	21 (61.8)	
**Continuous use of medications, n (%)**			
Yes	19 (32.2)	40 (67.8)	0.831 ^b^
No	15 (35.7)	27 (64.3)	
**Smoking, n (%)**			
Yes	5 (27.8)	13 (72.2)	0.784 ^b^
No	29 (34.9)	54 (65.1)	
**Family history of breast cancer, n (%)**			
Yes	14 (38.9)	22 (61.1)	0.510 ^b^
No	20 (30.8)	45 (69.2)	

^a^ Student’s *t*-test; ^b^ Fisher’s exact test. WCRF/AICR—World Cancer Research Fund/American Institute for Cancer Research. * 1st tertile represents the group with the lowest adherence to the WCRF/AICR recommendations. **Significant *p*-values are in bold.**

**Table 3 nutrients-17-01001-t003:** Cox proportional hazards regression models analyses of clinical outcomes of women with breast cancer according to score tertiles of adherence to 2018 recommendations of the WCRF/AICR (1st tertile versus 2nd and 3rd tertiles). Florianopolis, Santa Catarina, Brazil, 2025.

Variable	Unadjusted Analysis ^a^	Adjusted Analysis ^b^			Adjusted Analysis ^b,c^	
	HR (CI 95%)	*p*	HR (CI 95%)	*p*	HR (CI 95%)	*p*
**Overall mortality** **(n = 24 *)**	2.02 (0.71–5.74)	0.185	1.12 (0.47–2.66)	0.782	1.75 (0.54–5.69)	0.348
**Breast cancer-specific mortality** **(n = 15 *)**	1.19 (0.37–3.81)	0.254	0.89 (0.21–3.73)	0.875	1.53 (0.30–7.81)	0.606
**Overall 10-year survival ^d^ (n = 84 *)**	0.16 (0.03–0.8)	**0.025**	0.16 (0.28–0.96)	**0.045**	0.17 (0.02–1.2)	0.076
**Recurrence** **(n = 36 *)**	0.91 (0.43–1.91)	0.810	0.97 (0.45–2.1)	0.946	0.97 (0.45–2.1)	0.946
**10-year recurrence ^e^** **(n = 24 *)**	1.32 (0.48–3.63)	0.580	1.43 (0.48–4.23)	0.510	1.43 (0.48–4.23)	0.510

^a^ Cox proportional hazards regression models (crude); ^b^ Cox proportional hazards regression models adjusted for age, education, stage of first cancer, and family history of cancer. ^c^ Overall mortality, breast cancer-specific mortality and overall 10-year survival were also adjusted for recurrence. ^d^ Overall 10-year survival refers to women who did not die within 10 years of the first diagnosis of breast cancer. ^e^ The term “10-year recurrence” refers to recurrence within 10 years of the primary diagnosis of breast cancer. Group in the 1st tertile of adherence to the WCRF/AICR recommendations was considered the reference group for the regression models and represents the group with the lowest adherence score to the WCRF/AICR recommendations. * Numbers of events for each outcome. CI 95%, 95% confidence interval. HR—hazard ratio; WCRF/AICR—World Cancer Research Fund/American Institute for Cancer Research. **Significant *p*-values are in bold.**

## Data Availability

The original contributions presented in the study are included in the article and Appendix A; further inquiries can be directed to the corresponding author.

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
