# Peer review of "Lower Adherence to Lifestyle Recommendations of the World Cancer Research Fund/American Institute for Cancer Research (2018) Is Associated with Decreased Overall 10-Year Survival in Women with Breast Cancer"

_nutrients, 2025, doi:10.3390/nu17061001_

Round 1
Reviewer 1 Report
Comments and Suggestions for Authors
The topic covered in the manuscript deserves much consideration, and since the size of the sample observed is small, it is necessary to take good care of the text to allow its publication in a scientific journal.
The Authors should review the bibliographical references (for example reference 14 does not support what is stated in the text).
In describing the characteristics of the sample it is necessary to use also the same categorization used in the figures, informing readers with the estimates found in the set of observations of the 2nd and 3rd tertile.
Figures 3s and 4s should be published in the text, rather than in the supplementary material.
The data shown in tables S2, S3, S4, in which there are no significant associations between event, tertile of the WCRF score, and other characteristics detected in the sample highlight the problem of the small sample size indicated above.
The authors should have preliminarily calculated the size of the sample to be observed to highlight the relationship between eating habits and events in women with breast cancer.
The evaluation of adherence to lifestyle advice in a sample of patients to evaluate end-points such as survival from the disease or overall survival implies that the sample size is determined as well as the effort made in collecting the data. I appreciate the effort made in a single center, but it could be useful to aggregate data from multiple clinical centers, evaluate patients' conditions with the aid of pathology registers, recognize that for breast cancer survival shows a progressive improvement over the years, compared to which the difficulties in detecting specific benefits also from lifestyle measures increase.
Author Response
|
Thank you very much for taking the time to review this manuscript. Please find the detailed responses below and the corresponding revisions. All text adjustments and additions to the manuscript have been highlighted in green (Word version).
|
||||||||||||||||||||||||||||||||||||||||
|
3. Point-by-point response to comments and suggestions for authors |
||||||||||||||||||||||||||||||||||||||||
|
Comment 1: The topic covered in the manuscript deserves much consideration, and since the size of the sample observed is small, it is necessary to take good care of the text to allow its publication in a scientific journal.
Response 1: We appreciate your comment and understand that the sample size was indeed smaller than we expected. However, as stated on page 03, line 100, the sample was obtained for convenience. We had 136 participants at the beginning of the study but, due to the long research period, we lost 36 women to follow-up.
Additionally, we addressed this limitation in the study discussion:
“Regarding the limitations of the study, it is recognized the difficulty in accessing the information of many patients who had to be excluded from the sample. This was partly due to the long period of patient follow-up, which resulted in a reduction in the research sample size.” (Page 14, lines 438-440).
We reinforce the importance of the study and highlight that, despite having a small sample size, it is the first research in Brazil involving the analysis of survival and the relationship with following the WCRF/AICR recommendations in women with breast cancer. The long period of follow-up of the participants, exceeding 10 years, adds robustness to the study.
Comment 2: The Authors should review the bibliographical references (for example reference 14 does not support what is stated in the text).
Response 2: We appreciate your comment and have reviewed the references in our manuscript. The reference indicated (14) has been rectified:
Previous reference: Turati F, Bravi F, Polesel J, Bosetti C, Negri E, Garavello W. et al. Adherence to the Mediterranean diet and nasopharyngeal cancer risk in Italy. Cancer Causes Control. 28, 89-95 (2017). doi: 10.1007/s10552-017-0850-x.
Adjusted reference: Turati F, Dalmartello M, Bravi F, Serraino D, Augustin L, Giacosa, A. et al. Adherence to the World Cancer Research Fund/American Institute For Cancer Research Recommendations and the risk of breast cancer. Nutrients, 2020, 12, 607. doi: 10.3390/nu12030607.
Comment 3: In describing the characteristics of the sample it is necessary to use also the same categorization used in the figures, informing readers with the estimates found in the set of observations of the 2nd and 3rd tertile.
Response 3: We agree with your comment and thank you for your careful observation. In response to your suggestion, we reformulated Table 2 referring to the 2 groups: 1st tertle; and 2nd and 3rd tertile, as described in the rest of the text (page 8):
Table 2. Sociodemographic and clinical characteristics of breast cancer women according to the score tertiles of adherence to the WCRF/AICR recommendations. Florianopolis, Santa Catarina, Brazil, 2025.
aStudent's T-Test; bFisher's exact test. WCRF/AICR, World Cancer Research Fund/American Institute for Cancer Research. *1st tertile represents the group with the lowest adherence to the WCRF/AICR recommendations. Comment 4: Figures 3s and 4s should be published in the text, rather than in the supplementary material.
Response 4: We appreciate your suggestion and have incorporated the supplementary figures into the manuscript file (pages 9–11).
Comment 5: The data shown in tables S2, S3, S4, in which there are no significant associations between event, tertile of the WCRF score, and other characteristics detected in the sample highlight the problem of the small sample size indicated above. The authors should have preliminarily calculated the size of the sample to be observed to highlight the relationship between eating habits and events in women with breast cancer.
The evaluation of adherence to lifestyle advice in a sample of patients to evaluate end-points such as survival from the disease or overall survival implies that the sample size is determined as well as the effort made in collecting the data. I appreciate the effort made in a single center, but it could be useful to aggregate data from multiple clinical centers, evaluate patients' conditions with the aid of pathology registers, recognize that for breast cancer survival shows a progressive improvement over the years, compared to which the difficulties in detecting specific benefits also from lifestyle measures increase.
Response 5: We acknowledge that the study's sample size did not reach the desired level; however, we emphasize that it was a convenience sample. Despite this limitation, we recognize the study's value as an important precursor to future research on this topic among women with breast cancer in Brazil. We have highlighted this sample size limitation in our manuscript:
“Regarding the limitations of the study, it is recognized the difficulty in accessing the information of many patients who had to be excluded from the sample. This was partly due to the long period of patient follow-up, which resulted in a reduction in the research sample size.” (Page 14, lines 438-440).
In relation to the data presented in tables S2, S3 and S4, we understand that in fact the sample size may have impacted fewer statistically significant associations, but we still had important associations between daily consumption of sugary drinks and overall mortality (p = 0.005, table S2); daily fiber intake and overall 10-year survival (p = 0.027, table s2); daily consumption of sugary drinks and overall mortality (p = 0.026 in the adjusted model, table s3); daily consumption of fruits and vegetables and breast cancer-specific mortality (p = 0.022 in the adjusted model, table s3); daily fiber intake and overall 10-year survival (p = 0.022 in the adjusted model, table s3). We consider that these findings from the present study can provide scientific support for new studies with larger populations within the Brazilian context. |
||||||||||||||||||||||||||||||||||||||||
|
|
||||||||||||||||||||||||||||||||||||||||
|
4. Response to Comments on the Quality of English Language |
||||||||||||||||||||||||||||||||||||||||
|
Point 1: The English is fine and does not require any improvement. |
||||||||||||||||||||||||||||||||||||||||
|
Response 1: We are glad to know that the English is fine.
|
||||||||||||||||||||||||||||||||||||||||
Thank you very much for all your considerations to improve the manuscript.
Reviewer 2 Report
Comments and Suggestions for Authors
Lower adherence to lifestyle recommendations of the World Cancer Research Fund/American Institute for Cancer Research (2018) is associated with decreased overall 10-year survival in breast cancer women
This article addresses a very relevant public health issue, the influence of lifestyle on breast cancer survival, which is the most frequent cancer (after skin cancers) in females.
- The authors have indicated the volumes of WCRF/AICR that report the recommendations. For readers, it could be useful to have also a reference to the documents accessible in electronic format
- Very minor changes: line 70, put 5 in brackets
- CUP released a report that synthesizes the latest evidence on diet, nutrition, physical activity and body weight providing guidance for people living with and beyond breast cancer. Authors should refer to this report which condenses many studies on the topic [https://www.wcrf.org/research-policy/evidence-for-our-recommendations/after-a-cancer-diagnosis-follow-recommendations/breast-cancer-survivors-research/]
- The authors used the full WCRF/AICR score (0-7 points). They could remind readers that an abbreviated score (0-5 points) is also used in the literature.
- The authors did not explain whether they had estimated the necessary sample size before proceeding with the investigation.
- The authors did not explicitly state the opinion of the ethics committee. They should also explain how informed consent was obtained if the interviews were by telephone.
- The study design deserves more detail. As far as we know, data collection was done between 2006 and 2011, so before the WCRF/AICR score was released. The first thing to clarify is this: was the score applied recently, based on data collected then? Were all the data complete?
- Second question on the method. Women who had a certain lifestyle between 2006 and 2011, therefore before the tumor, have maintained the same lifestyle also after the tumor? None of them changed their WCRF/AICR score from 2006 to 2025?
- It does not appear that the study can be classified as longitudinal. It is a retrospective study, in which archival information was retrieved. It is not clear whether the authors verified that the habits recorded many years ago were maintained until today and how they behaved with those who changed habits.
- The above points explain the difficulty of evaluating the study. It certainly has the merit of having addressed a very important topic. However, there are doubts about the power of the method and the size of the sample, probably too small to fully observe the phenomenon. Any change in habits could have induced an unevaluated confounding factor.
Author Response
|
Comment 1: The authors have indicated the volumes of WCRF/AICR that report the recommendations. For readers, it could be useful to have also a reference to the documents accessible in electronic format.
Response 1: We appreciate your comment and agree with your suggestion. Therefore, we include links to the electronic formats of the WCRF/AICR reports in the references:
5. World Cancer Research Fund/American Institute for Cancer Research. Diet, Nutrition, Physical Activity and Cancer: a Global Perspective. Continuous Update Project Expert Report, 3rd ed, 2018; pp. 4-112. Available online: https://www.wcrf.org/wp-content/uploads/2024/11/Summary-of-Third-Expert-Report-2018.pdf (accessed on 26 February 2025).
6. World Cancer Research Fund/American Institute for Cancer Research. Food, nutrition, physical activity, and the prevention of cancer: a global perspective. Project Expert Report, Washington, DC, 2nd ed. Washington, DC, 2007; pp. 3-13. Available online: https://discovery.ucl.ac.uk/id/eprint/4841/1/4841.pdf (accessed on 26 February 2025).
Comment 2: Very minor changes: line 70, put 5 in brackets
Response 2: We appreciate your comment and added the brackets:
Considering the WCRF/AICR recommendations released in 2007 [6], which present some differences from the updated recommendations [5], (…).
Comment 3: CUP released a report that synthesizes the latest evidence on diet, nutrition, physical activity and body weight providing guidance for people living with and beyond breast cancer. Authors should refer to this report which condenses many studies on the topic [https://www.wcrf.org/research-policy/evidence-for-our-recommendations/after-a-cancer-diagnosis-follow-recommendations/breast-cancer-survivors-research/]
Response 3: Your suggestion is great and was included in the text of the manuscript:
Page 2, lines 43-48:
This report had a previous version published in 2007 [6], and in 2024, the new WCRF/AICR report synthesized the latest evidence on lifestyle factors to provide guidance for individuals living with and beyond breast cancer. Key health components, such as maintaining a healthy body weight and consuming adequate dietary fiber, have been associated with improved health outcomes [7].
Reference included:
7. World Cancer Research Fund/American Institute for Cancer Research. Diet, nutrition, physical activity and body weight for people living with and beyond breast cancer. Cancer Update Programme, 1st ed, 2024; 1-50. Available online: https://www.wcrf.org/wp-content/uploads/2025/01/CUP-Global-BCS-Report.pdf (accessed on 26 February 2025).
Comment 4: The authors used the full WCRF/AICR score (0-7 points). They could remind readers that an abbreviated score (0-5 points) is also used in the literature.
Response 4: It is indeed important to include this information in the text, and we appreciate your suggestion. The following text has been added to the manuscript:
We would also like to highlight that the most recent abbreviated version of the score [44] has already been released. However, we chose to use the proposal by Shams-White et al. [43], as it incorporates more components of the WCRF/AICR guidelines.
Reference included: 44. Malcomson FC, Parra-Soto S, Ho FK, Celis-Morales C, Sharp L, Mathers JC. Abbreviated score to assess adherence to the 2018 WCRF/AICR Cancer Prevention Recommendations and risk of cancer in the UK Biobank. Cancer Epidemiology, Biomarkers & Prevention, 2024, 33, 33-42. doi: 10.1158/1055-9965.EPI-23-0923
Comment 5: The authors did not explain whether they had estimated the necessary sample size before proceeding with the investigation.
Response 5: We appreciate your comment and understand that the sample size was indeed smaller than we expected. However, as stated on page 03, line 100, the sample was obtained for convenience. We had 136 participants at the beginning of the study but, due to the long research period, we lost 36 women to follow-up.
Additionally, we addressed this limitation in the study’s discussion:
“Regarding the limitations of the study, it is recognized the difficulty in accessing the information of many patients who had to be excluded from the sample. This was partly due to the long period of patient follow-up, which resulted in a reduction in the research sample size.” (Page 14, lines 438-440).
We reinforce the importance of the study and highlight that, despite having a small sample size, it is the first research in Brazil involving the analysis of survival and the relationship with following the WCRF/AICR recommendations in women with breast cancer. The long period of follow-up of the participants, exceeding 10 years, adds robustness to the study.
Comment 6: The authors did not explicitly state the opinion of the ethics committee. They should also explain how informed consent was obtained if the interviews were by telephone.
Response 6: We appreciate your suggestions and have incorporated more information into the text to better clarify the ethical procedures of the research as follows in the text below (page 7, lines 236-257):
2.4 Ethical approval The present study followed the ethical principles of research and was approved by the Ethics Committee for Research with Human Beings of the Federal University of Santa Catarina (099/2008, 492/2009 and 4.536.968), Ethics Committee of the Oncological Research Center (015/2009, 4.639.465, 5.148.664) and Ethics Committee of the Carmela Dutra Maternity Hospital (0012.0.233.242-10). The study was only conducted with patients (or family members, in the case of deceased patients) who signed an online version of the Free and Informed Consent Form, as established in Resolution 466/2012 of the National Health Council [65]. The Free and Informed Consent Form was sent in the format of an online form (Google Forms) to patients and/or their families via a link provided via WhatsApp and/or email. It should be noted that the Ethics Committees were consulted regarding sending the Free and Informed Consent Form in digital format and approved this file format due to the COVID-19 pandemic. This form provided the necessary guidance for an adequate understanding of the research and was completed digitally by the participants and/or their families who were contacted via telephone call. A copy of the digitally completed Free and Informed Consent Form was sent to the email address of the form responder. Only individuals contacted by telephone who indicated in this document that they agreed to participate in the research took part in the study. The remaining women in the sample who were not contacted by telephone, and who already had all the necessary information in their physical and electronic medical records, did not require sending and signing an informed consent form, since data was obtained via clinical records. |
|
Comment 7: The study design deserves more detail. As far as we know, data collection was done between 2006 and 2011, so before the WCRF/AICR score was released. The first thing to clarify is this: was the score applied recently, based on data collected then? Were all the data complete?
Response 7: We agree with your point of view regarding the greater detail of study design. Therefore, we included the following text to clarify any doubts (page 4, lines 157-160):
The calculation of the WCRF/AICR score was carried out in 2021, that is, after data collection (baseline that occurred between 2006 and 2011). The research team had access to all the necessary data to enable the calculation of the WCRF/AICR score, without requiring the collection of any additional data for operationalization.
Comment 8: Second question on the method. Women who had a certain lifestyle between 2006 and 2011, therefore before the tumor, have maintained the same lifestyle also after the tumor? None of them changed their WCRF/AICR score from 2006 to 2025?
Response 8: Your questions are pertinent, and we would like to thank you for pointing out the information that should be better detailed in the manuscript. Therefore, we included the following text to clarify these questions (page 4, lines 151-156):
The participants' lifestyle was not reassessed in the second stage of data collection (out-comes data collection), since the aim of the research was to evaluate the influence of lifestyle at the time of breast cancer diagnosis on clinical outcomes of mortality, survival and recurrence. It is possible that the participants changed their lifestyle habits over the years of the study, however, the research focus consisted of assessing lifestyle at the baseline of clinical outcomes of interest to the research group.
Comment 9: It does not appear that the study can be classified as longitudinal. It is a retrospective study, in which archival information was retrieved. It is not clear whether the authors verified that the habits recorded many years ago were maintained until today and how they behaved with those who changed habits.
Response 9: Thank you for your considerations. Our study was classified in the manuscript as prospective and observational, since different types of variables were collected over time. For example, in the baseline, sociodemographic, clinical, anthropometric, physical activity and food consumption data were collected; Subsequently, in the “outcomes data collection”, data on mortality, survival and recurrence were obtained. In this way, the same variables were not collected over time, but the research was predominantly prospective.
To better clarify this issue for readers, we have included the following sentence in the manuscript (Page 3, lines 96-97):
We highlight that our study is classified as prospective and observational, since different types of variables were collected over time.
Comment 10: The above points explain the difficulty of evaluating the study. It certainly has the merit of having addressed a very important topic. However, there are doubts about the power of the method and the size of the sample, probably too small to fully observe the phenomenon. Any change in habits could have induced an unevaluated confounding factor.
Response 10: We sincerely appreciate your thorough review and valuable comments on our study. We acknowledge the challenges inherent in evaluating complex phenomena such as the impact of lifestyle habits on clinical outcomes in breast cancer.
Regarding the sample size, we emphasize that our study utilized a convenience sampling method, as described in the manuscript. While we recognize the inherent limitations of this approach, we believe that our findings provide meaningful insights into the relationship between lifestyle at the time of diagnosis and disease prognosis. Observational studies often face sample size constraints; however, they still offer valuable scientific findings, particularly when addressing critical clinical and public health issues.
Additionally, we understand the concern regarding potential confounding factors related to lifestyle changes over time. However, the scope of our study was specifically to assess lifestyle habits at the time of diagnosis and their association with mortality, survival, and recurrence. Therefore, reassessing habits at later stages was beyond the study’s objectives. This approach is justified by the potential impact of pre-diagnosis behaviors on clinical outcomes, aligning with prior research in the field.
We believe that our study makes a significant contribution by emphasizing the importance of evaluating lifestyle factors at the time of breast cancer diagnosis. We hope our responses clarify your concerns, and we remain open to any suggestions that may further enhance the manuscript.
|
|
4. Response to Comments on the Quality of English Language |
|
Point 1: The English is fine and does not require any improvement. |
|
Response 1: We are glad to know that the English is fine.
|
Thank you very much for all your considerations to improve the manuscript.
Reviewer 3 Report
Comments and Suggestions for Authors
After breast cancer diagnosis, survival is related to some main factors. Example: In hormone receptor negative, HER-2 positive cases, neoadjuvant treatment achieving pathologoanatomic complete response is associated with better survival. Please comment your findings with the above.
Author Response
|
Comment 1: After breast cancer diagnosis, survival is related to some main factors. Example: In hormone receptor negative, HER-2 positive cases, neoadjuvant treatment achieving pathologoanatomic complete response is associated with better survival. Please comment your findings with the above.
Response 1: We appreciate your comment and agree with your consideration. Therefore, we included the following text in the manuscript (Page 14, Lines 427-436):
When addressing the prognosis of patients with breast cancer, it is important to consider that there are factors not associated with lifestyle that can also influence survival. In hormone receptor-negative, HER2-positive breast cancer, achieving a pathologic complete response (pCR) after neoadjuvant therapy is strongly associated with improved survival outcomes, including higher overall survival and event-free survival rates. Studies have shown that patients who attain pCR have a significantly lower risk of recurrence and distant metastases compared to those with residual disease [66, 67]. A meta-analysis by Cortazar et al. [66] demonstrated that pCR is a robust surrogate marker for long-term survival, particularly in aggressive breast cancer subtypes, such as HER2-positive and triple-negative breast cancers. |
|
4. Response to Comments on the Quality of English Language |
|
Point 1: The English is fine and does not require any improvement. |
|
Response 1: We are glad to know that the English is fine.
Thank you very much for all your considerations to improve the manuscript.
|
Round 2
Reviewer 1 Report
Comments and Suggestions for Authors
I read the manuscript updated by the Authors who carefully followed most of the suggestions given previously. Reading the paper I have the impression that the Aauthors did not conceive the work described in the perspective of a scientific publication, but rather in the text there is a narration of their healthcare practice: in this sense it provides real-world evidence and I would have liked it if the Authors themselves underlined this limitation more forcefully.
Author Response
Dear Reviewer,
Thank you very much for your important contributions to my manuscript.
Please see the attachment.
Best regards,
Patricia Faria Di Pietro

Reviewer 2 Report
Comments and Suggestions for Authors
The authors have revised the manuscript
Author Response

(The authors gave the same response as above.)
